# Why Should This Article Be Deleted?
# Transparent Stance Detection in Multilingual Wikipedia Editor Discussions

**Lucie-Aimée Kaffee**[*]
Hasso Plattner Institute
Germany
lucie-aimee.kaffee@hpi.de

**Arnav Arora**[*]
University of Copenhagen
Denmark
aar@di.ku.dk

**Isabelle Augenstein**
University of Copenhagen
Denmark
augenstein@di.ku.dk

## Abstract

The moderation of content on online platforms is usually non-transparent. On Wikipedia, however, this discussion is carried out publicly and editors are encouraged to use the content moderation policies as explanations for making moderation decisions. Currently, only a few comments explicitly mention those policies – 20% of the English ones, but as few as 2% of the German and Turkish comments. To aid in this process of understanding how content is moderated, we construct a novel multilingual dataset of Wikipedia editor discussions along with their reasoning in three languages. The dataset contains the stances of the editors (*keep*, *delete*, *merge*, *comment*), along with the stated reason, and a content moderation policy, for each edit decision. We demonstrate that stance and corresponding reason (policy) can be predicted jointly with a high degree of accuracy, adding transparency to the decision-making process. We release both our joint prediction models and the multilingual content moderation dataset for further research on automated transparent content moderation.[1]

## 1 Introduction

Moderators typically discuss content moderation decisions they want to take collectively. In those discussions, it is crucial to understand the stances of the discussion participants. As moderators often refer to content moderation policies (Gillespie, 2018), these can be seen as a justification for their proposed solution for the problem at hand. Therefore, understanding the stance of moderators and how they come to a decision is crucial for community members to ensure that the decisions are made based on the policies the community has

---

[*] Both authors contributed equally to this work

[1]Code: ⓞ https://github.com/copenlu/wiki-stance
Dataset: 🤗 https://huggingface.co/datasets/copenlu/wiki-stance

agreed on. It has previously been utilised, using a wisdom-of-the-crowds approach, for moderation in specific scenarios, for instance, fact-checking (Hardalov et al., 2022b). Moderator decisions, such as changing the status of a subreddit to private in protest (Porter, 2023), is typically a decision made in dialogue with the community and based on their feedback and previously agreed upon policies for that community.

As stance detection is a crucial task for understanding the reasoning for content moderation decisions, we here propose a method to explain stances with existing moderation policies. Policies in general can take various shapes and forms (Robyn Caplan, 2018). For many online platforms, these policies are hidden in Terms and Conditions documents and point to what is permissible or not on a platform, typically to mitigate potential harms (Arora et al., 2023). For others, they are explicitly stated for the community to refer to, where these policies should be included in the considerations when content is discussed (Gillespie, 2018). Communities on those platforms further have norms and policies of their own (Fiesler et al., 2018). For members of such communities, it is imperative to be able to comprehend why their content was deleted based on these policies. For the platforms themselves, highlighting such policies are crucial for maintaining the trust and safety of the platform and is also required by upcoming regulation (Kaminski and Urban, 2021). Finally, for the content moderators on those platforms, navigating through a large number of legalese-heavy policies can be challenging and time-consuming. However, to the best of our knowledge, supporting content moderators with automated tools that jointly predict a moderator's decision and the corresponding policy has not been explored in prior research.

Such a discussion process referring to policies exists, for example, on Wikipedia, a community-edited encyclopedia. Here, policies ensure the qual-

ity of the written articles and that standards are kept. For example, the *Manual of Writing* describes how an article should be written, the *Reliable sources* policy ensures that enough and high-quality references are included in an article, and the *Notability* policy ensures that only topics notable enough for an encyclopedic entry are included on Wikipedia. Some form of these policies exists across different language Wikipedias, ensuring that the respective community standards in each language are clearly communicated and upheld.

If an article does not live up to the communities' standards, editors on Wikipedia can recommend the article for deletion. These deletion discussions are a form of content moderation process, in which editors cite established policies to argue for the deletion of, e.g., a group or post. Across all Wikipedia language versions, the process is very similar: An article is suggested for deletion, a *Deletion Discussion* takes place in which the editors *support* or *oppose* the deletion of the article, or propose to *merge* it with other articles or *comment* on the discussion. For an example comment in such a deletion discussion, see Figure 1. Eventually, an admin on Wikipedia takes action using the input of the discussion. A variety of guidelines are developed to help editors make this decision.[2]

When discussing the deletion of articles, many editors refer to existing policies in Wikipedia, to base their arguments on these evidence documents. In discussions, referring to evidence documents is helpful, as it grounds arguments in more objective points and previously agreed-upon standards (Xiao and Askin, 2014; Schneider et al., 2012). Automating the task of getting an overview and proposing policies to cite is crucial to help admins make consistent decisions, and for users to understand how and based on which policies decisions are discussed. Currently, there is limited work in the area of transparent content moderation, and none of it has considered policies in community discussions as explanations or reasoning for stance. We therefore propose a multi-task setup in which stance is detected alongside the prediction of a justification in the form of a policy, as described in Figure 2. As different language Wikipedias have different sizes in terms of articles and accordingly deletion discussions, we further propose a multilingual setup, aligning policies across different

---

[2]See, for example, deletion policy on English Wikipedia: https://en.wikipedia.org/wiki/Wikipedia:Deletion_policy

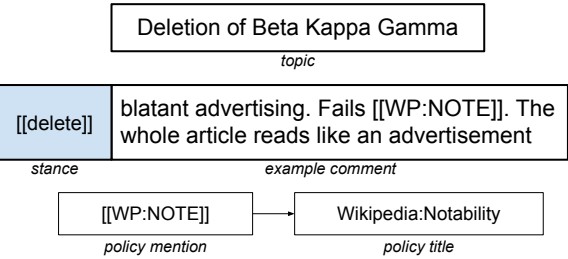

Figure 1: Example comment from the dataset

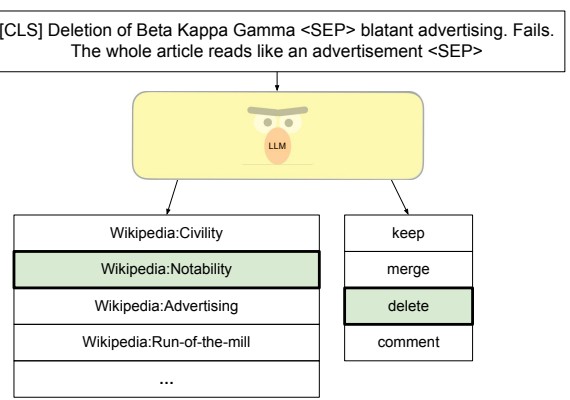

Figure 2: Overview of the approach for policy prediction and stance detection

language Wikipedias.

To this end, our contributions are as follows:

- A transparent stance dataset in three languages (English, German, and Turkish), annotated with stance as suggested by the editor, as well as policies as arguments for their stance, and a multilingual version in which policies are aligned across languages;

- Benchmarking of Transformer-based models on this multilingual dataset;

- An approach which leverages a multi-task setup to predict stance along with a policy as a justification for the stance;

- Empirically demonstrated improvements of this multi-task setup over the single task setup for low-resource scenarios (Turkish).

## 2 Related Work

**Policies for Content Moderation**   Online content moderation and, with it, the policies that moderators use are indispensable on the current online platforms (Langvardt, 2017; Gillespie, 2018). Kittur and Kraut (2010) describe that once a community grows, its implicit rules need to be formalised

into standards, guidelines, or policies. Social media platforms, such as Facebook, have policies to aid content moderators in deleting or keeping posts of users. Conflict arises when those policies are applied non-transparently (Sablosky, 2021).

As content moderation is a crucial topic, there have been proposals to automate aspects of it. Risch and Krestel (2018) propose an automated method to suggest the deletion of social media comments based on general guidelines, such as *insults, discrimination, and defamation*. They do however not analyse the discussion of content moderators themselves. Ribeiro et al. (2023) suggest that automated content moderation could increase adherence to community standards. In our paper, we focus on the policies used in Wikipedia. They are created and maintained by community members, as any Wikipedia page can be edited by any user. These policies play an important role in negotiating community standards and have a wide range of functions (Butler et al., 2008). To the best of our knowledge, there is no prior work on automatically detecting policies in content moderator discussions as a justification for their decisions.

**Deletion Discussion on Wikipedia** Xiao and Askin (2014); Schneider et al. (2012) find that in deletion discussions, rationales stated by the editors are rooted in established policies. One of the important aspects for editors contributing to the deletion discussions is the familiarity with the norms of these discussions (Geiger and Ford, 2011), in which citing existing policies can be of great help.

The deletion discussions have been analysed in an automated manner to understand how online communities communicate. Xiao and Sitaula (2018) detect sentiment in the deletion discussions on English Wikipedia and find that positive sentiments tend to indicate an article being kept, and negative sentiments tend to indicate that the article will be deleted. Mayfield and Black (2019b,a) model Wikipedia deletion discussions as group decision-making processes. To this end, they use stance detection to evaluate each user's attitude. They show the importance of understanding stance in the Wikipedia deletion discussions. However, they only work on a subset of the deletion discussions in the English Wikipedia. Park et al. (2015) propose a method to identify online user arguments in need of support, but do not take the next step of proposing supporting documents.

**Transparent Stance Detection** Stance detection is a well-studied task, with prior work covering different ways of formalising it, considering a variety of different targets and forms of text, as well as different domains (Hardalov et al., 2021, 2022a,b).

Less well-studied, however, is explainable stance detection. Providing a reason for a stance prediction is important to make it comprehensible and transparent to users why that decision was made (Draws et al., 2023). One approach to make stance more explainable is to have human-annotated rationales for stance detection in a training dataset, which a model can then learn to predict for unseen text (Jayaram and Allaway, 2021). Another approach is to use a topic modeling approach to extract stance labels alongside textual justifications for these labels. (Gómez-Suta et al., 2023) However, no existing approaches have leveraged policies to explain the stance labels predicted, despite these being natural explanations of stance in content moderation discussions.

## 3 Transparent Stance Detection

In content moderation, obtaining an overview of the attitudes of different moderators towards a decision is crucial to reach a consensus. The reasoning of the proposed decisions by moderators is ideally grounded in agreed upon community standards. In Wikipedia, deletion discussions are a forum for exchanges about whether an article should be deleted or kept on the platform. In our work, we aim to support the admins, who make the final decision on whether an article should be deleted or kept, by automating the process of getting an overview of the different attitudes. Given a comment on a topic, such as the example comment in Figure 1, we aim to predict the stance of the user along with their reasoning in form of a policy, as displayed in Figure 2. The task of transparent stance detection consists of two subtasks: 1) policy prediction, in which we predict the reasoning for a decision in the form of a policy; and 2) stance detection, in which the stance of the comment is detected. Transparent stance detection aims to predict the policy and stance of a comment alongside each other. We explore these tasks in a multilingual manner, leveraging knowledge from high-resource language Wikipedias to improve task performance on lower resourced ones.

**Policy Prediction** In Wikipedia deletion discussions, rationales for opinions should be rooted in the policies on Wikipedia, and mostly are, even if

only implicitly (Xiao and Askin, 2014; Schneider et al., 2012). Referring to the policy documents directly in discussions can be helpful for other users to understand the reasoning easily. Still, a large part of the comments do not directly refer to a policy on Wikipedia (see Section 4). For our task of transparent stance detection, the policy document can be a justification for the decision (stance) the moderator proposes in their comment. Therefore, we predict the relevant Wikipedia policy given an editor's comment. This could be extended to comments that do not yet refer to a policy and support editors and moderators to gain an easy overview over opinions and their rationales. In our paper, we leverage the fact that the policies are used as justification to explain the stance of the user, by using it for transparent stance detection (see Section 3).

**Stance Detection**   Stance Detection has historically been used for understanding discussions at large. It allows us to understand an individual's position towards a topic. We define stance along the commonly adopted definition in the field of natural language processing, as the expression of a speaker's standpoint and judgement towards a given proposition (Biber and Finegan, 1988). In deletion discussions, the stances of users are given as votes to decide whether an article should be kept or deleted, i.e. the proposition is the deletion of a given article, and the standpoint is expressed through *keep*, *delete*, *merge*, or *comment* suggestions (which we use as stance labels) that the author argues for in their comment. Any editor can contribute to these discussions. Automating the detection of stances in moderators' comments can be helpful to content moderation at large, as it allows moderators to get a high-level overview of the opinions voiced in a discussion. Prior work has suggested the same for detecting the stances of news articles so that fact-checkers can quickly get a quick overview and take an informed judgement about the veracity of an article on a topic (Pomerleau and Rao, 2017). Based on a user's comment and the name of the article discussed, we predict the stance of the comment towards the deletion or retention of the article.

**Transparent Stance Detection**   For users of a platform, it is crucial to understand why content moderation decisions are made. Our paper contributes to the field of stance detection in content moderation by jointly considering the tasks of pol-

| Language | train | test | dev | #pol |
|---|---|---|---|---|
| en | 372,033 | 43,776 | 21,961 | 94 |
| de | 7,320 | 862 | 455 | 48 |
| tr | 684 | 202 | 44 | 33 |

Table 1: Dataset split statistics

| Label | #instances | | |
|---|---|---|---|
| | en | de | tr |
| delete | 279063 | 4805 | 433 |
| keep | 108273 | 3394 | 252 |
| comment | 30974 | 395 | 224 |
| merge | 19460 | 43 | 21 |

Table 2: Distribution of stance labels in the dataset

icy prediction and stance detection as part of a larger framework, thereby creating a transparent stance detection setup. In this transparent stance setup, given a comment and a topic, we predict the stance alongside the policy, justifying the stance. In content moderation, this setup can be used to understand and summarise the attitudes of different moderators and clarify which community standards or policies form the basis of a moderation decision.

**Multilingual Transparent Stance Detection**
Different Wikipedia languages drastically vary in size in terms of the number of articles as well as editors. Each of the different Wikipedia language communities creates policies so as to have standards to point new editors to. Across Wikipedias, cultural norms and standards may differ (Hara et al., 2010), but there is still a high overlap in the guidelines they write. As such, different Wikipedia language versions can benefit from each other in terms of identifying new guidelines to add or old ones to adapt. This is particularly true for lower resourced Wikipedias, in which there are fewer editors to maintain the communities' policies and fill gaps when a new policy should be created.

## 4   Dataset

We create three new datasets, containing annotations of stances and policies in articles for deletion discussions in the English, German, and Turkish Wikipedias. The datasets cover the time span from 2005 (2006 for Turkish) to 2022. Each instance in the dataset consists of the title of the page that is discussed for deletion as the topic, the stance towards

the topic, i.e. deletion of the article, the comment of the user arguing for or against deletion, and the policy they cite to justify their argument. An example comment, shown in Figure 1, is "*[[delete]] blatant advertising. Fails [[WP:NOTE]]. The whole article reads like an advertisement*", where *delete* is the stance of the editor and the mentioned policy is *Notability*. More examples from the dataset can be found in the Appendix in Table 5.

## 4.1 Dataset Creation

To create the dataset, we identify the article deletion discussion archive pages for the English, German, and Turkish Wikipedia respectively, and retrieve all deletion discussions in the considered time frame through the respective MediaWiki APIs[3]. From those pages, we select comments which mention a Wikipedia page, identified by the prefix `[[WP:` or `[[Wikipedia:`. We find that these generally refer to policies, or policy abbreviations such as *WP:NOTE* in our example. If the policy abbreviations link to a policy page, the Wikimedia API resolves them and returns the actual policy or Wikipedia page title. For each of the three languages, we retrieve the full policy page through the Wikimedia API, manually select the policies that are actual policy pages, and discard other Wikipedia pages, such as articles. We further discard all policies that are mentioned infrequently (100 in English, 10 for German, and 2 for Turkish, due to the varying data set sizes) across all comments in the respective language deletion discussions. The final policies include, for example, the notability criteria in the English Wikipedia[4], in German Relevanzkriterien[5], in Turkish kayda değerlik[6] To collapse sub-polices with the same or similar meaning, or subcategories of one policy into the main policy, we merge them based on the link of the sub-policy to the main policy in the policy page text, e.g., notability criteria for specific article types such as *Wikipedia:Notability (music)* were merged into the *Wikipedia:Notability* policy. This was done manually based on the original as well as machine translated versions of the policy texts by an annotator proficient in German and English with basic understanding of Turkish. The number of policies in our dataset varies considerably by language, see Table 1. As the majority of comments refer to only one policy, we keep only one policy per comment by selecting the first policy mentioned. We further remove all mentions of policies from the comments using regular expressions, which often breaks grammaticality of the sentence but is necessary to prevent leakage of label information. For future work, we also release a unperturbed version of the dataset, with the policy mentions intact. The stance labels (*keep*, *delete*, *merge*, and *comment*), can be expressed in different forms or spelled differently. We manually identify the different ways the labels might be expressed and aggregate them into the four standard labels. In the Appendix, Table 2, we list all versions of each of the labels across the three languages considered. To maintain privacy, and since this information is not relevant to our experiments, we remove usernames from the comments using regular expressions. The dataset is split in train/test/dev, where the split for English and German is 80%/15%/5%, but due to the low number in comments in Turkish, we decided to alter the split for Turkish to have at least 200 test examples. We use this dataset for the task of policy prediction and stance detection, as well as transparent stance, in which we predict stance alongside policy, see Section 6.

**Label Analysis** We sampled a subset of 100 randomly selected comments from the English test dataset for a preliminary human annotation study on this subset, for both stance and policy prediction. One annotator familiar with the English Wikipedia read the comments as present in the dataset, i.e., where stance and policy are removed, and then, for each comment, selected the matching stance and policy from the list of 94 English Wikipedia policies. The accuracy scores between our annotation and the labels (which were self annotated by the Wikipedia editors) are 88% and 55% for stance and policy prediction respectively. We attribute the relatively lower score for policy prediction to the large number of labels (94), i.e., policies, with a random prediction baseline of 1%. These policies also at times overlap with each other, e.g., *Wikipedia:Lists* and *Wikipedia:Categories, lists, and navigation templates*. Further, the comments are at times quite short, making it challenging for the annotator to select the correct policy.

---

[3]English Wikipedia API, German Wikipedia API, Turkish Wikipedia API

[4]https://en.wikipedia.org/wiki/Wikipedia:Notability

[5]https://de.wikipedia.org/wiki/Wikipedia:Relevanzkriterien

[6]https://tr.wikipedia.org/wiki/Vikipedi:Kayda_de%C4%9Ferlik

## 4.2 Multilingual Dataset

Since Wikipedia policies are connected across languages by links, we can leverage these cross-language links to reuse the mentions of the same policies in other languages' comments for languages with lower resources. Non-English language Wikipedias may have fewer articles and therefore fewer discussions overall, but also, crucially for this paper, the use of policies might be limited as fewer policies exist. Aligning and predicting policies across languages has the advantage that a cross-lingual model can implicitly learn further relationships between policies, and thereby achieve an improved performance for target languages at test time. In future work, this might enable Wikipedias with fewer policy pages to automatically find gaps in which policies already existing in other language Wikipedias they might want to establish.

We then create a dataset, which concatenates Turkish, German, and English for training and validation based on the datasets introduced in Section 4.1. The comments in the test data are the same as in the original test data for the three languages, i.e., we provide a test dataset for each of the three languages. The stance detection labels are the same for all languages and can thus easily be mapped to one another. For policies, we create a super set across policies present in all 3 languages. In this super set of policies, we kept the policies intact that only exist in one language, and merged the policies that exist across two or all three languages. To merge the policies, we use the fact that the policy pages link to each other on Wikipedia (inter-language links), in the same way different language Wikipedia articles on the same topic link to each other. When we merge the policies, we refer to the English title in the other languages. This super set of policies finally contains 116 policies across three languages by linking a total of 63 English and German policies to English. Hence, when performing policy prediction for Turkish, a model can leverage the presence of those policies in English or German.

## 4.3 Dataset Statistics

The number of mentioned policies is heavily skewed towards the notability policy in each language, with 56% of comments mentioning the notability policy in English, 45% of comments in German, and 59% of comments in Turkish, which

makes it important to ensure that models do not overfit when predicting policies. Comments, after removing policies and other information as described above, have on average 248.3 (English), 342.3 (German), and 381.7 (Turkish) characters. There is a high variance between the languages regarding how often policies are referred to in the discussions: 21.4% of English comments refer to a policy, however only 2.7% of German, and 2.2% of Turkish comments refer to a policy in the observed time frame.This explains the small size of the dataset for these languages, but also shows the need for automating the references to policies, as proposed in Section 3.

## 5 Experimental Setup

We consider four setups for our evaluation of individual and joint prediction of stance and policy: i) *Single task* where we consider the tasks outlined in Section 3 individually while only using the training set for that language; ii) *Multi-task* where we do joint prediction of both the tasks, while only using the dataset for the corresponding language; iii) *Single task Multilingual*, where we train on the multilingual dataset for all languages but test on a single language for one task; and iv) *Multi-task multilingual*, where we train on the multilingual dataset and test on a single language, predicting both stance and policy jointly. We use three types of Transformer models for our experiments, BERT (Devlin et al., 2019), mBERT (Devlin et al., 2019), and XLM-R (Conneau et al., 2020). For each of the setups, we run a learning rate sweep between [5e-4, 5e-5, 5e-6], taking the best model from each. We train the models with a batch size of 8, optimizing for accuracy on the development set during policy prediction and macro-averaged F1 on the development set during stance detection or joint prediction. For the single language experiments, we use the language specific version of BERT, mBERT, and XLM-R, whereas for the multilingual experiments, we only use the multilingual models mBERT and XLM-R. The models were run for 5 epochs for the English and multilingual setups and 500 epochs for the smaller German and Turkish datasets. To avoid overfitting, we use an early stopping mechanism based on the aforementioned evaluation metric with a patience of 5. We report macro-averaged F1 scores for the task of stance detection as it is a 4-way classification task with labels skewed towards the *delete* label. For policy prediction, however,

| | single task | | | multi-task | | | multiling. single task | | | multiling. multi-task | | |
|---|---|---|---|---|---|---|---|---|---|---|---|---|
| | en | de | tr | en | de | tr | en | de | tr | en | de | tr |
| Random | 0.20 | 0.19 | 0.17 | – | – | – | – | – | – | – | – | – |
| Majority | 0.19 | 0.18 | 0.15 | – | – | – | – | – | – | – | – | – |
| lang_BERT | 0.80 | 0.51 | 0.48 | 0.80 | 0.45 | 0.54 | – | – | – | – | – | – |
| XLMR | 0.79 | 0.48 | 0.48 | 0.79 | 0.40 | 0.50 | 0.78 | 0.43 | 0.51 | 0.77 | 0.30 | 0.31 |
| mBERT | 0.78 | 0.50 | 0.41 | 0.77 | 0.37 | 0.45 | 0.79 | 0.41 | 0.48 | 0.76 | 0.38 | 0.47 |

Table 3: Macro-averaged F1 scores for **stance detection**, for the English, German, and Turkish datasets for different setups; single task, multi-task, and both setups for the multilingual dataset. For the monolingual models (*lang_BERT*), we use BERT, pre-trained on English, German, and Turkish data respectively. The best result across all setups for each language is underlined.

| | single task | | | multi-task | | | multiling. single task | | | multiling. multi-task | | |
|---|---|---|---|---|---|---|---|---|---|---|---|---|
| | en | de | tr | en | de | tr | en | de | tr | en | de | tr |
| Random | 0.01 | 0.02 | 0.06 | – | – | – | – | – | – | – | – | – |
| Majority | 0.55 | 0.45 | 0.62 | – | – | – | – | – | – | – | – | – |
| lng_BERT | 0.76 | 0.72 | 0.69 | 0.72 | 0.61 | 0.68 | – | – | – | – | – | – |
| XLMR | 0.77 | 0.68 | 0.63 | 0.68 | 0.53 | 0.63 | 0.76 | 0.67 | 0.74 | 0.63 | 0.46 | 0.60 |
| mBERT | 0.75 | 0.75 | 0.66 | 0.67 | 0.56 | 0.65 | 0.75 | 0.68 | 0.70 | 0.66 | 0.51 | 0.60 |

Table 4: Accuracy scores for **policy prediction**, for the English, German, and Turkish datasets for different setups; single task, multi-task, and both setups for the multilingual dataset. For the monolingual models (*lang_BERT*), we use BERT, pre-trained on English, German, and Turkish data respectively. The best result across all setups for each language is underlined.

the number of labels are substantially higher, making per-label examples scarce, especially for the smaller datasets. Hence, in this scenario, we report accuracy scores.

**Baselines**   To ground the results, we provide a *Random* baseline and a *Majority Class* baseline for the single task setups. The former is calculated by assigning a random label from the label set to each test instance while the latter is calculated based on distribution of the labels in the test set.

**Single task setup**   For the single task setup, we use the standard fine-tuning procedure for the models. We take the [CLS] representation of the combined topic and comment text and fine-tune a sequence classification head to make the predictions for stance and policy respectively.

**Multi-task setup**   To make a joint prediction, we use a multi-task learning framework (MTL) with hard parameter sharing (Caruana, 1993) as shown in Figure 2. This is inspired by prior work which successfully employs MTL for joint classification and explanation generation (Atanasova et al., 2020).

In this setting, both tasks are learnt jointly, allowing the model to leverage important label information across the tasks. In order to further enhance this sharing, we freeze the first 6 layers of the encoder for each model. Moreover, to stabilise the losses and balance the learning between the two tasks, we alternate between the losses in the ratio 3:1 for stance detection and policy prediction every step. Empirically, we find that this allows for the models' losses to converge for both tasks without the models overfitting.

**Multilingual setup**   Since the German and Turkish Wikipedias are smaller than the English one, the datasets reflecting their deletion discussions contain fewer instances as well. To leverage shared learning of the task across languages, in this setup, we train the models on a joint set of comments from the English, German, and Turkish datasets and tested on a single language test set. While stance labels are common (with translated labels) across the three languages, policies across each language Wikipedia differ, making it challenging to consolidate them to a single set of labels. To

circumvent this, we manually align the Turkish and German policies with the English policies. As policies can differ across languages, there is not an exact match for all policies. However, largely the policies do align across languages, see Section 4.2.

# 6 Results

In the following, we present the results for the experiments we conducted in order to transparently detect a stance of a users comment, i.e., detect stance alongside the policy used as a justification for the stance.

## 6.1 Single Task

For the tasks of policy prediction and stance detection, individually trained and tested, we find that the models perform reasonably well compared to the majority and random baselines for both tasks. The results for the policy prediction task can be found in Table 4, and the results for the stance detection task in Table 3, under the *single task* column. Between the languages, we can see the impact of the different dataset sizes, especially for the stance detection task. Here, all three models trained on the smaller German and Turkish datasets perform substantially worse compared to ones trained on the English set. However, policy prediction is a task where models for all three languages achieve comparable results, suggesting that seeing more instances does not necessarily help in being able to predict the relevant policy for a comment, where the number of labels is much larger. We analyse the most common bi-grams using LIME, and find that the most salient features learned are largely different lexicalisation of the labels, e.g., *not enough* for the delete class (see Appendix B.1). For the monolingual English BERT model *comment* is most commonly confused with *delete*, based on its confusion matrix(see Appendix B.2). The overall scores are promising and suggest that the tasks of stance detection and policy prediction on the deletion discussion comments are feasible to automate.

## 6.2 Multi-task

To support content moderators and give users insights into both the stances of editors and relevant content policies in a discussion, we propose to predict stances and policies jointly by combing the two tasks in a multi-task model setup (Section 5). While the setting is more challenging, we find that multi-task models can achieve comparable scores

to single-task ones, see *multi-task* results in Table 4 and 3. For Turkish, the performance improves compared to the single task setup for stance detection. This demonstrates that identifying the relevant content policy for a comment can also be beneficial for the predicting the stance of a comment, especially for an extreme low resource setting. For German, however, the policy prediction task suffers a drop in performance in this setting.

## 6.3 Multilingual learning

In the experiments on the joint dataset of all languages (Tables 3 and 4), under the *multilingual single-task* column, we find similar results as for the multi-task setting. In the very low resource scenario of Turkish, we see an improvement in performance across both stance detection and policy prediction. However, for German, there is a performance drop across both tasks. The results for English remain consistent in this setting.

For the *multilingual multi-task* setup, the combined superset of aligned policies is 116, higher than policy labels for any one language, making this setup more challenging and not directly comparable to the other *single task* and *multi-task* settings for policy prediction. The scores for policy prediction paint a similar picture, with a drop in performance compared to the other settings. For stance as well, contrary to prior results in the literature, where cross-lingual learning has substantially improved performance on the task (Hardalov et al., 2022a), we find that learning from high resource languages does not aid in our multi-task setting.

# 7 Discussion and Conclusion

Making content moderation decisions more transparent is crucial for users to understand why decisions were ultimately made and to understand how they were discussed. Prior work has shown that explanations for content removal are underutilised in content moderation. Adding them leads to more user engagement, reducing the odds of future post removals (Jhaver et al., 2019). Moreover, a mandate of providing users with an explanation for why their post or content was deleted is also included in upcoming regulation (Kaminski and Urban, 2021). Thus, making content moderation more transparent and explainable is beneficial to both online platforms and end users.

To aid in this moderation, we need improved NLP methods for stance detection. However, re-

sources for NLP across languages are heavily skewed (Joshi et al., 2020). This is especially the case for stance detection where multilingual resources are scarce (Hardalov et al., 2022b). In an effort to address these issues, we create a multilingual dataset (consisting of three languages: English, German, and Turkish) based on Wikipedia deletion discussions, annotated with the stance of the discussion participant and the policy they refer to in their comment as a justification. Given a comment in a content moderation discussion, we predict the stance of the moderator alongside a justification in the form of a policy.

For an end-user of Wikipedia, the joint prediction setup proposed here is particularly insightful as they get a high level overview of the votes for or against deletion of an article along with the policy cited as justification for the stance. For Wikipedia editors themselves, such a setup can help establish good practices of citing Wikipedia content policies in their comments potential candidates provided by the model. The setup can also help make a speedy decision through a bird's-eye view of the discussion with preservation of some of the nuance beyond just *keep* or *delete* labels. Our results show that it is possible to learn the stance and policy prediction tasks jointly, aiding in the transparency of automatic content moderation models. For Turkish, he multi-task setup outperforms the monolingual setup for stance, demonstrating that leveraging the information across the two tasks can benefit their predictions for low-resource settings. As Wikipedias across languages vary in size in terms of articles and therefore discussion pages, we also experiment with a multilingual setup to support Wikipedia languages with fewer data resources. To this end, we align policies across languages where possible. We find that for Turkish, the smallest dataset of our experiments, this multilingual training can improve performance, especially for the policy prediction task. As there are fewer policy pages overall for Turkish and German, such a setup could in the future also be used to predict policies missing in one language that exist already in another Wikipedia language. Community norms vary and it is important to allow for their diversity to exist while still making use of information from norms of other communities. Our setup allows for such learning while still supporting the editors in making informed decisions based on the editor discussions. In the future, the task of transparent

stance detection should be applied and tested on more platforms to encourage more transparent and comprehensible content moderation. As policies for content moderation differ across platforms, our model is not able to directly predict other platforms' policies in a zero-shot setting. However, the model could be fine-tuned on another platforms' policies and potentially leverage the policy-content relationship observed on Wikipedia We believe that the general framework that we propose as well as the challenges faced in defining the problem as a classification task has implications beyond Wikipedia, to most platforms where norms or policies are explicitly defined. We would also look to further investigate the arguments used within the comments, along with the policies mentioned, to increase the transparency and potentially model performance across the tasks.

## Acknowledgements

This research was partially funded by a DFF Sapere Aude research leader grant under grant agreement No 0171-00034B, as well as by the Pioneer Centre for AI, DNRF grant number P1.

## Limitations

One natural limitation of our approach is the fact that not all data of deletion discussions can be leveraged as not all comments refer to policies (see Section 4). This severely limits the size of our dataset. Further, in the multilingual setup, we might predict policies for, e.g., Turkish or German, which are matching the comment's content but do not exist in the language Wikipedia's policies yet.

## Ethics Statement

As the dataset is real data, taken from Wikipedia deletion discussions, it is important to handle it with care; there might be sensitive content in the data and one should not use it to identify individual editors. The latter we addressed by removing usernames wherever possible. For more details on the dataset, we provide the datasheet for datasets, see Appendix C.

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

## A  Dataset Creation

As the decision of each moderator is written by hand, labels with the same meaning were merged into one of the four main labels by language. Comments without a decision label and comments which were ambiguously labelled were discarded. In Table 2, we detail which strings were merged into which labels.

## B  Model Analysis

To get a better understanding of the dataset, model behaviour, and its failure cases, we conduct analysis of the English subset of the data outlined below.

### B.1  Most Salient Features

To get a sense of what are the most salient words corresponding to each of the labels, we train a logistic regression classifier using default parameters on TF-IDF features for the English training set. Using LIME (Ribeiro et al., 2016), we then plot the most salient bi-grams associated with each label as shown in Figure 4. We can see that negative expressions like *fails and* and *not enough* are very highly weighted with the delete class and negatively weighted for the keep class, which makes intuitive sense. Similarly, *clearly passes* and *easily passes*, which are presumably highlighting a policy that the article fulfils are associated with the keep class, further demonstrating the intertwined nature of editor stance and policy references. The bi-gram *per nom*, an abbreviation for *per nominator* or *per nomination*, has positive weights for the delete and merge class while negative ones for the other two, indicating that when the article's nomination for deletion is referenced, the article tends to be voted

| Topic | Comment | Stance | Policy |
|---|---|---|---|
| Deletion of Scottish Nursery Nurses Strike | per #NEWS. No source to establish lasting significance of this particular strike | delete | What Wikipedia is not |
| Deletion of Motivational press | Fails by a wide margin. I see a lot of article mentioning the publisher but on taking a closer look, they are all trivial mentions. In fact some of them are made by a certain Justin Sachs who works for the company. I had a look at the article creator's talk page | delete | Notability |
| Deletion of V. T. Rajshekar | per with major award noted above | keep | Notability |
| Deletion of Sabeetha Wanniarachchi | The beauty contests she has taken part in are not major, and I couldn't find any significant coverage in reliable sources about her. She does not satisfy the as a result | delete | Notability |
| Deletion of No Mountains in Manhattan | the album has received coverage at [Link] Pitchfork, [Link]The Fader, [Link]Exclaim!, and [Link] XXL, all of which are considered reliable sources at our Wikipedia:WikiProject Albums/Sources\|Albums WikiProject. Additional coverage exists at [Link] Village Voice and [Link] BrooklynVegan. With these in mind, the album meets and | keep | Notability |
| Löschung von Fleur Klingelberger | 1. Zwischeparken bis Erreichen der \|Relevanz, z.Z. im ANR | delete | Notability |
| Löschung von Rhein-Münsterland-Express | Relevanzkriterien http://de.wikipedia.org/wiki/WP:RK bei Verkehrswege und -bauwerke enthalten nach meiner Lesart "Eisenbahnstrecke". Generelle Diskussionen sollten in anderem Rahmen geführt werden. Sehe aber das Dilemma welches uns hier wieder einmal vor die Füße wirft | keep | Do not disrupt Wikipedia to illustrate a point |
| Löschung von COVID-19-Pandemie/Statistik | Die Daten sind international nicht vergleichbar und von miserabeler Datenqualität; selbst wenn man die deutlich besseren Daten von Johns Hopkins University nehmen würde, ist das schlicht Schrott gemäß . Also . Ich weiß, wird nicht passieren, weil ist ja belegt!!!!!einself | delete | No original research |
| Löschung von Gustav Wohlgemuth | Habe den Artikel ein bisschen überarbeitet, mit Quellen und einem Literaturhinweis versehen. Wohlgemuth war Professor in Leipzig und erfüllt schon deshalb die #Wissenschaftler\|Relevanzkriterien. Deshalb (und natürlich auch, weil ich das Ding nicht umsonst bearbeitet haben will :-D | keep | Notability |
| Löschung von Nagelpflegemittel | Einträge in zwei anerkannten Lexika, daher ist #Allgemeine Anhaltspunkte für Relevanz\|Relevanz gegeben. LAE Fall 1: Die Begründung des Löschantrags trifft eindeutig nicht oder nicht mehr zu | keep | Notability |
| Sigortam.net'in silinmesi | Yukarıda "\|İlgili kayda değerlik kriterlerimizin ikinci maddesi (Site veya sitedeki içerik, bir kurum veya yayın organının verdiği, iyi bilinen ve bağımsız bir ödül almış olmalı) gereğince KD kabul edilmiyor mu?" şeklinde bir yorum yapmıştım. Bu kriteri karşıladığından kalmalı madde | keep | Notability |
| Kurtuluş Yolu'nun silinmesi | KD olduğunu düşünen arkadaşlar, bu kanıya hangi kriterleri göz önüne alarak vardıklarını belirtebilir mi? İlgili kriterler \|burada | comment | Notability |
| Ferec'in silinmesi | Ferec grubu \|kayda değerlik 1 ve 6'ya net bir şekilde uyuyor. 4'üncü maddeye de uyması için bir kaç ay var | keep | Notability |
| Emin Atlı'nın silinmesi | M6'dan direkt silinebilir. kriterlerini karşılamıyor | delete | Notability |
| Zekeriya Önge'nin silinmesi | bu varsa şu da olmalıdır diye değil, 'deki kriterlere uygun olup olmadığına bakılarak karar veriliyor. –Kullanıcı mesaj:Kibele\|"'kibele | delete | Notability |

Table 5: Dataset examples for the three languages. *Topic* is the target of the stance detection, *Comment* is where the stance itself is expressed by the Wikipedia editor. Links in the table are replaced with [Link] for readability

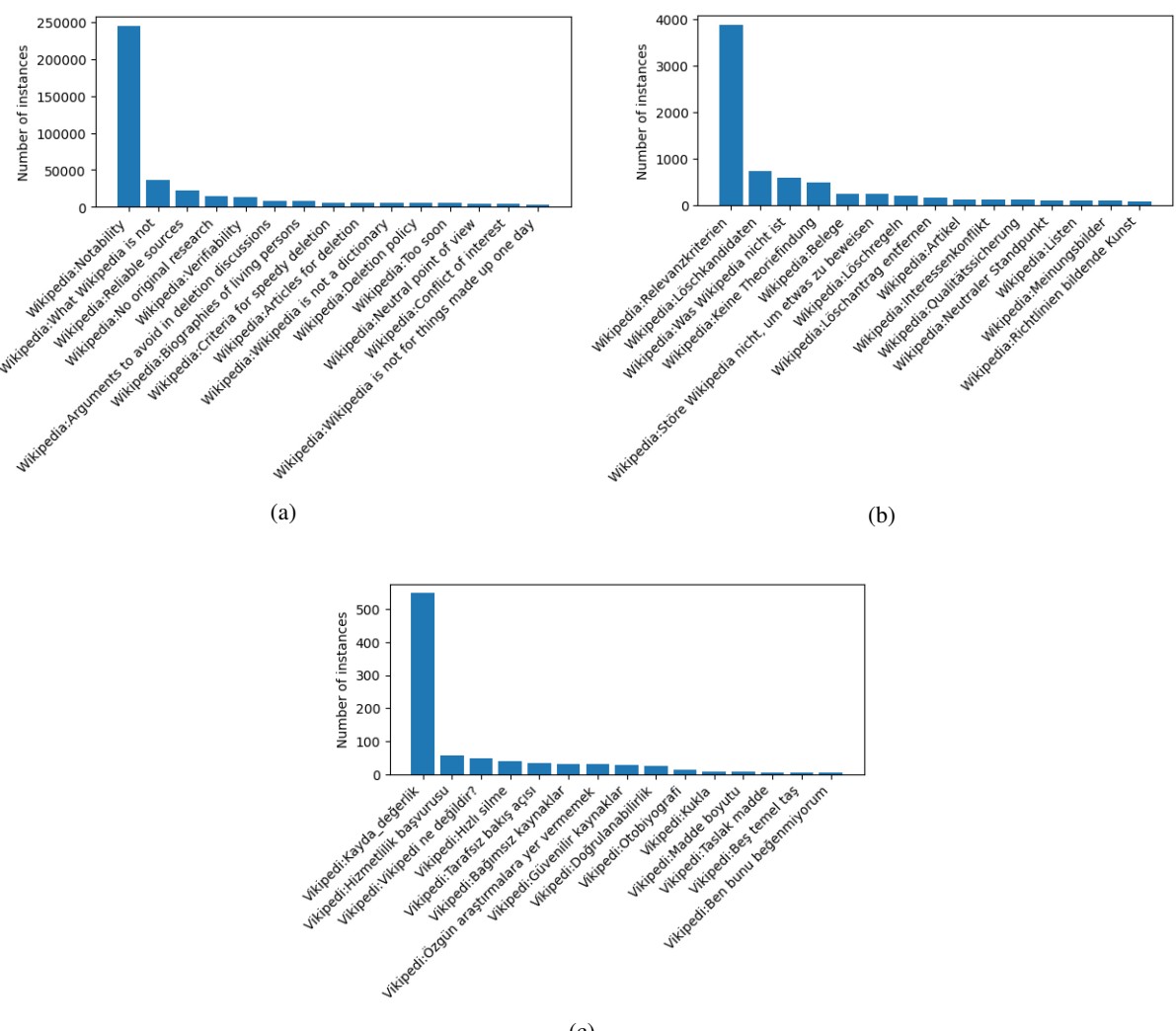

Figure 3: The 15 most frequently used policies across comments for (a) English, (b) German, and (c) Turkish.

for deletion or merging. The bi-gram with the highest weight is *can you*, which is intuitive since a request for a change is being made.

## B.2 Error Analysis

We performed a brief error analysis of the best model for stance detection task in English i.e. English BERT. Below, we provide the confusion matrix (Table 7), F1 scores per label (Table 8) as well as examples of comments which the model got wrong with the highest probability (Table 9). Looking at the confusion matrix and F1 scores per label, the *comment* category is clearly the category the model has the hardest time with, often being mixed up with *keep* or *delete*. This makes intuitive sense as comments conceptually capture a broader range of statements than the other categories. The model has a tendency to over-predict *delete* due to its over-representation in the dataset, leaving scope for future studies to address this skewness. Looking at examples of erroneous predictions, we can see instances of the most frequent errors (*delete-keep*, *delete-comment*) made by the model.

## C Datasheet for Datasets

In the following, we provide the Datasheet for Datasets as described by Gebru et al. (2021).

### C.1 Motivation

**For what purpose was the dataset created? Was there a specific task in mind?** The aim of the dataset is to support automated content moderation by providing justification for a decision of a moderator as well as to predict the decision based on the comment.

**Who created the dataset (e.g., which team, research group) and on behalf of which entity (e.g., company, institution, organization)?** The researchers listed as authors from their respective institutions.

**Who funded the creation of the dataset?** See acknowledgements.

### C.2 Composition

**What do the instances that comprise the dataset represent (e.g., documents, photos, people, countries)?** The dataset contains a Wikipedia editor's comment in a deletion discussion, along with their decision between *delete*, *keep*, *merge*, or *comment*. The stance labels are merged from the larger labels, e.g., *speedy keep* is merged into the decision

for *keep*. For the comments, the topic in the form of the article discussed for deletion is provided. Further, each comment is annotated with one policy they mention in the decision. We provide two datasets, one in which the comment is left as it was provided by the editor and one where all policies are removed from the comment. We provide the dataset for three different language Wikipedias; English, German, and Turkish.

**How many instances are there in total (of each type, if appropriate)?** The dataset consists of 437,770 comments for English, 8,637 comments for German, and 930 comments for Turkish, split into train, test, and dev. The split for Turkish is not in the same proportion as for the other languages, to ensure a reasonable amount of examples in the test set.

**Does the dataset contain all possible instances or is it a sample (not necessarily random) of instances from a larger set?** The dataset is limited to comments, that contain policies. The full dataset of all comments with stance annotations contains 2,047,698 comments for English, 319,576 comments for German, and 43,043 comments for Turkish. This means, our dataset contains 21.4% of English, 2.7% of German, and 2.2% of the comments of the overall dataset.

**What data does each instance consist of?** The comments are processed to remove policies and are annotated with one policy per comment along the merged stance labels. For the topics, we added *Deletion of* to the article title.

**Is there a label or target associated with each instance?** The labels are taken from the comment itself, i.e., the first policy mentioned in the comment along the stance label provided by the editor. The stance labels are processed to fit into the four categories of *delete*, *keep*, *merge*, or *comment*. Further, the name of the article that is discussed for deletion is provided.

**Is any information missing from individual instances?** Where possible, we removed the time stamp and mentions of usernames. The timestamps were removed as they are not relevant to the task, and the usernames were removed where possible for anonymity. As this is done automatically using regex, there might be non-standard ways of mentioning usernames or times, that could not be removed

| y=comment top features | | y=delete top features | | y=keep top features | | y=merge top features | |
|---|---|---|---|---|---|---|---|
| Weight? | Feature | Weight? | Feature | Weight? | Feature | Weight? | Feature |
| +4.743 | can you | +9.081 | fails and | +6.904 | clearly passes | +9.225 | to list |
| +4.458 | my nomination | +7.951 | per nom | +6.321 | easily passes | +8.107 | or delete |
| +3.989 | this discussion | +6.685 | non notable | +6.097 | clearly notable | +6.799 | search term |
| +3.986 | you are | +6.265 | no reliable | … 118638 more positive … | | +5.690 | per nom |
| +3.984 | not sure | +6.091 | not enough | … 125579 more negative … | | +5.574 | is blp |
| +3.977 | this afd | +5.913 | no evidence | -6.118 | not notable | +5.481 | standalone article |
| … 105779 more positive … | | +5.328 | no sources | -6.236 | non notable | +5.326 | the redirect |
| … 138438 more negative … | | +5.232 | fails the | -6.279 | no reliable | +5.230 | as suggested |
| -4.617 | per above | +5.131 | not notable | -6.389 | none of | +5.206 | independent notability |
| -4.747 | per the | … 111174 more positive … | | -7.252 | per nom | +5.038 | to the |
| -5.877 | per and | … 133043 more negative … | | -9.100 | not enough | … 73651 more positive … | |
| -6.390 | per nom | -5.246 | or delete | -10.010 | fails and | … 170566 more negative … | |

Figure 4: Most salient bi-grams for each label in the training set

**Are relationships between individual instances made explicit (e.g., users' movie ratings, social network links)?** Different comments are connected by the topic they discuss, i.e., the title of the article that is under discussion for deletion.

**Are there recommended data splits (e.g., training, development/validation, testing)?** The train/test/dev splits are provided. They are standard splits of 80/15/5 for English and German. Due to the small dataset size of the Turkish dataset, we decided to alter the split so that there is a reasonable number of comments in the test set (at least 200 comments).

**Are there any errors, sources of noise, or redundancies in the dataset?** As we automatically process real user comments, errors are unavoidable, but to the best of our knowledge, we limited them to a minimum. There might be minor errors in the stance labels. Further, as we are working with only the first policy mentioned, there might be important policy mentions that are overlooked. However, as the median number of policies mentioned for all three datasets is 1, we believe that this will have very little impact on the tasks working with policies.

**Is the dataset self-contained, or does it link to or otherwise rely on external resources (e.g., websites, tweets, other datasets)?** The dataset is self-contained, no other resources are needed to work with it for the described tasks.

**Does the dataset contain data that might be considered confidential (e.g., data that is protected by legal privilege or by doctor–patient confidentiality, data that includes the content of individuals' non-public communications)?** As it is real data, it is possible that data is included that falls under these concerns. However, as the data is public and comments can be edited by the editors after posting, we believe that this data should be free of confidential information.

**Does the dataset contain data that, if viewed directly, might be offensive, insulting, threatening, or might otherwise cause anxiety?** As deletion discussion data contains interactions of people on different language Wikipedias, we cannot ensure that there is no offensive data, and we would like this to be considered for future reuse. Particularly, as articles of real people might be under discussion of deletion, this data should be handled with care.

**Does the dataset identify any subpopulations (e.g., by age, gender)?** The dataset removes usernames where possible. Where usernames might have been overlooked, we would like to point out that this is how the editors want to identify and is not necessarily connected to their real-life identity.

**Is it possible to identify individuals (i.e., one or more natural persons), either directly or indirectly (i.e., in combination with other data) from the dataset?** As above, we remove usernames where possible. If usernames are accidentally included in the dataset, it is only possible to identify the user on Wikipedia.

**Does the dataset contain data that might be considered sensitive in any way (e.g., data that reveals race or ethnic origins, sexual orientations, religious beliefs, political opinions or union**

memberships, or locations; financial or health data; biometric or genetic data; forms of government identification, such as social security numbers; criminal history)?   Only data from the discussion itself is included, which should not be able to identify any sensitive characteristics of any of the editors contributing to these conversations.

### C.3   Collection Process

**How was the data associated with each instance acquired?**   The data was scraped from the public archive of deletion discussions of Wikipedia in English, German, and Turkish. This data can be accessed at English, German, Turkish.

**What mechanisms or procedures were used to collect the data (e.g., hardware apparatuses or sensors, manual human curation, software programs, software APIs)?**   We collected the topics and comments in an automated manner from the discussion pages. We then processed the stance labels by semi-automatically merging them into the four stance labels described above. The policies were automatically extracted by looking for the `[[Wikipedia:` (or `[[Vikipedi:` for Turkish) prefix. As this leaves us with a large number of links that might contain also other Wikipedia pages, such as WikiProjects, we manually processed the policies by merging them and removing comments linking to irrelevant pages. We further manually edited the Turkish topics to contain the equivalent of *Deletion of* in Turkish, as vowel harmony made it difficult to automatically add this. For English and German *Deletion of* and *Löschung von* respectively were added automatically. The alignment of policies across the three languages was done manually, relying on the Wikipedia interwiki links, which link Wikipedia pages across languages.

**If the dataset is a sample from a larger set, what was the sampling strategy (e.g., deterministic, probabilistic with specific sampling probabilities)?**   We only retain comments containing stances and policies. This sample might not be representative of the overall deletion discussion but was necessary for the task.

**Who was involved in the data collection process (e.g., students, crowdworkers, contractors) and how were they compensated (e.g., how much were crowdworkers paid)?**   The manual editing of, e.g., the policies, the editing of the Turkish topics, and the linking policies across languages, was conducted by the authors.

**Over what timeframe was the data collected?**   The dataset was collected over the course of a few days at the end of 2022, and contains data from 2005 (2006 for Turkish) until 2022.

**Were any ethical review processes conducted (e.g., by an institutional review board)?**   No ethical review process was conducted.

**Did you collect the data from the individuals in question directly, or obtain it via third parties or other sources (e.g., websites)?**   The data was collected from the English, German, and Turkish Wikipedia deletion discussion pages (see above for the links).

**Were the individuals in question notified about the data collection?**   It was not possible to contact individuals, as the data was on Wikipedia, we could not notify each user about the collection of the data.

**Did the individuals in question consent to the collection and use of their data?**   N/A

**If consent was obtained, were the consenting individuals provided with a mechanism to revoke their consent in the future or certain uses?**   N/A

**Has an analysis of the potential impact of the dataset and its use on data subjects (e.g., a data protection impact analysis) been conducted?**   As we removed usernames where possible, we believe the dataset should not have a direct impact on the data subjects.

### C.4   Preprocessing/cleaning/labeling

**Was any preprocessing/cleaning/labeling of the data done (e.g., discretization or bucketing, tokenization, part-of-speech tagging, SIFT feature extraction, removal of instances, processing of missing values)?**   The usernames, timestamps, and policies were removed from the comments.

**Was the "raw" data saved in addition to the preprocessed/cleaned/labeled data (e.g., to support unanticipated future uses)?**   We provide a version of the dataset in which the comment contains the policies. We do not provide version with usernames and timestamps. This data can be found "raw" on the Wikipedia discussion pages linked above.

**Is the software that was used to preprocess/clean/label the data available?** https://github.com/copenlu/wiki-stance

## C.5 Uses

**Has the dataset been used for any tasks already?** The dataset has been used for the tasks described in this paper.

**Is there a repository that links to any or all papers or systems that use the dataset?** We aim to list them on the GitHub or HuggingFace dataset page.

**What (other) tasks could the dataset be used for?** The dataset could be used for further experiments in the field of content moderation.

**Is there anything about the composition of the dataset or the way it was collected and preprocessed/cleaned/labeled that might impact future uses?** The described removal of usernames, timestamps, and policies should be considered in the future use of the dataset.

**Are there tasks for which the dataset should not be used?** Any work related to identifying Wikipedia users is discouraged.

## C.6 Distribution

**Will the dataset be distributed to third parties outside of the entity (e.g., company, institution, organization) on behalf of which the dataset was created?** The dataset will be freely accessible under a permissive license.

**How will the dataset will be distributed (e.g., tarball on website, API, GitHub)?** The dataset will be freely available on GitHub and at a common website for dataset distribution.

**When will the dataset be distributed?** The dataset will be distributed after acceptance of this paper.

**Will the dataset be distributed under a copyright or other intellectual property (IP) license, and/or under applicable terms of use (ToU)?** As the text is derived from Wikipedia, and Wikipedia is licensed under the creative commons licenses (CC-BY-SA) and GNU Free Documentation License (GFDL), we will license the dataset under a similar, compatible dataset license.

**Have any third parties imposed IP-based or other restrictions on the data associated with the instances?** Wikipedia data is licensed under CC-BY-SA and GFDL, which has to be respected for further distribution of the data.

**Do any export controls or other regulatory restrictions apply to the dataset or to individual instances?** Wikipedia data is licensed under CC-BY-SA and GFDL, which has to be respected for further distribution of the data.

## C.7 Maintenance

**Who will be supporting/hosting/maintaining the dataset?** The authors can be contacted for the maintenance of the dataset.

**How can the owner/curator/manager of the dataset be contacted (e.g., email address)?** The first authors as listed above can be contacted.

**Is there an erratum?** No.

**Will the dataset be updated (e.g., to correct labeling errors, add new instances, delete instances)?** As of now, there are no plans to update the dataset beside correcting errors in the existing version of the dataset. However, it could be recreated for newer deletion discussions.

**If the dataset relates to people, are there applicable limits on the retention of the data associated with the instances (e.g., were the individuals in question told that their data would be retained for a fixed period of time and then deleted)?** N/A

**Will older versions of the dataset continue to be supported/hosted/maintained?** If errors are detected in the dataset, e.g., missed usernames in the automatic processing of the comments, the authors are committed to fixing these errors.

**If others want to extend/augment/build on/contribute to the dataset, is there a mechanism for them to do so?** The dataset will be available on HuggingFace datasets(https://huggingface.co/datasets/copenlu/wiki-stance), where issues can be opened to notify the authors. The authors can also be contacted via email.

| Label | Label in comment |
|---|---|
| **English** | |
| keep | keep, oppose, save, retain, keeep, preserve, stay |
| delete | delete, support, agree, del |
| comment | comment, note, question, reply, response, clarification, answer, request, query, neutral, abstain, commmment, undecided, uncertain, not sure, unsure, no opinion, coment, fyi, no vote |
| merge | merge, redirect, move, rename, rewrite, split, relist |
| **German** | |
| keep | behalten, lae, bleibt, relevant, bleiben, nein, behalt, erhalten |
| delete | löschen, gelöscht, sla, lösch, løschen, loeschen, weg, wech, entsorgen, hinfort |
| comment | 7 tage, sieben tage, neutral, überarbeiten, unentschieden |
| merge | verschieben, redirect, weiterleitung |
| **Turkish** | |
| keep | kalsın, kalsin, silinmesin, kalması |
| delete | silinsin, silinmeli, sil |
| comment | yorum, çekimser, cevap, soru, tarafsız, ping, seslen, mesaj, düzenle, yanıt, kararsız, geçersiz |
| merge | aktarılsın, birleştirilsin, aktarılması, taşınsın |

Table 6: Labels for each language and text representations of each of these labels that were merged into the labels, across the three languages

| | F1 score |
|---|---|
| comment | 0.47 |
| delete | 0.94 |
| keep | 0.89 |
| merge | 0.76 |

Table 8: Macro-averaged F1 scores per label for English BERT in the single task setting

| | comment | delete | keep | merge |
|---|---|---|---|---|
| comment | 1083 | 1121 | 866 | 22 |
| delete | 188 | 27092 | 498 | 112 |
| keep | 182 | 795 | 9830 | 51 |
| merge | 19 | 456 | 150 | 1311 |

Table 7: Confusion Matrix for Stance Detection with BERT in English

| Instance | Predicted Label | Actual Label |
|---|---|---|
| no notablity proven, the references do not mention the subject of the article and only one, at best, is a reliable source | delete | comment |
| and redirect, fails , and . The term is a neologism and is not used in a manner to denote fiscal policy, either contemporarily or in archaic terminology. — | delete | keep |
| fails on all counts. Also potential | delete | comment |
| The sources added by voceditenore seem to adequately establish the notability requirements at | delete | keep |
| Seems like a notable article. It needs a ǀreliable source though | delete | comment |
| as it appears to exist for the purpose of a ǀsynthetic association between this scene and the TV pilot | keep | merge |
| "Fails as no sources discuss this particular data set. Only references are GRG lists (note that none of these sources discuss ""Caribbean supercentenarians"" and are just a big table of names) which do nothing to establish notability. All names in this article are available in other Longevity articles" | delete | keep |
| Per , all arguments for deletion are based on subjective reasons. All the information comes from Time magazine New York Times, and others prime sources | delete | keep |
| "as not-notable under guidelines, as well as and concerns. If author(s) can cite ǀverifiable ǀsources that this, ""Has received a major award for excellence in some aspect of filmmaking,"" or ""Has been the subject of multiple, non-trivial news stories describing its artistic or societal impact"" I would consider it notable" | delete | comment |
| Speculative and highly inaccurate cherry-picked . No place for this ruminative dissertation on Wikipedia | delete | keep |
| "the coverage is routine and / or ; a wholly unremarkable, minor chain. If promotionalism is removed (""...with a vision of ""spreading the vegetarian lifestyle worldwide...!""), there won't be anything left. Wikipedia is not a directory of nn businesses, nor is it an opportunity to place franchisee ads" | delete | comment |

Table 9: Most confident erroneous predictions by the English BERT model for stance detection