# OpenReview forum: "Why Should This Article Be Deleted? Transparent Stance Detection in Multilingual Wikipedia Editor Discussions"
_EMNLP/2023/Conference — EMNLP 2023 Main_

### Official Review · Reviewer_Xa7D · 2023-08-05

**Soundness:** 4

**Excitement:**

4: Strong: This paper deepens the understanding of some phenomenon or lowers the barriers to an existing research direction.

**Paper Topic And Main Contributions:**

The paper focuses on developing models for predicting the stance and policy (reason) for Wikipedia content moderation in three languages.

**Reasons To Accept:**

- Addresses the important issue of content moderation (a very important problem), with a focus on transparency.
- Shows successful application to Wikipedia editor discussions in three languages, highlighting the potential for multilingual applicability.
- The release of the multilingual dataset and joint prediction models could foster more research and development in content moderation and transparency.

**Reasons To Reject:**

- Limited applicability due to the focus on Wikipedia, which might make the method too niche. It would help if the authors point out other applicable areas.
- Related with the above, It is uncertain whether the method can be generalized to other out-of-distribution problems or online platforms other than Wikipedia. Further exploration and validation on other platforms would strengthen the paper.
- Unclear if the F1 scores between 0.6 - 0.8 have a significant real-world impact.


**Reproducibility:**

4: Could mostly reproduce the results, but there may be some variation because of sample variance or minor variations in their interpretation of the protocol or method.

**Reviewer Confidence:**

3: Pretty sure, but there's a chance I missed something. Although I have a good feel for this area in general, I did not carefully check the paper's details, e.g., the math, experimental design, or novelty.

---

> ### Author Rebuttal · Authors · 2023-08-28
>
> We thank the reviewer for their appreciation of transparent content moderation as a valuable and important topic.
>
> Many online platforms, such as Reddit or Facebook, set overall content policies [1][2] that communication on the platform has to adhere to. Communities on those platforms further have norms and policies of their own. For members of such communities, it is imperative to be able to comprehend why their content was deleted based on these policies. For the platforms themselves, highlighting such policies are crucial for maintaining the trust and safety of the platform and is also required by upcoming regulation. Finally, for the content moderators on those platforms, navigating through a large number of legalese heavy policies can be challenging and time consuming. Hence, joint moderation and linking the relevant policies has demonstrable impact and can make the process tenable for the moderators on all platforms.
>
> However, to the best of our knowledge, joint prediction of the moderation decision and the corresponding policy has not been explored in prior research. As policies for content moderation differ across platforms, our model wouldn’t be able to predict other platforms’ policies in a zero-shot setting. However:
> * The model could be fine-tuned on another platforms’ policies and potentially leverage the policy-content relationship observed on Wikipedia
> * We believe that the general framework that we propose as well as the challenges faced in defining the problem as a classification task (for instance, overlap of policies) has implications beyond Wikipedia, to most platforms where norms or policies are explicitly defined.
>
> To address the reviewer’s point of F1 scores of 0.6-0.8 not having real world impact: the purpose of our study is not to provide a cross-domain tool for transparent content moderation. Rather, it is to present research on a first attempt to conceptualise and address this novel task, with Wikipedia as an example, that we hope the research community can improve upon and the larger community can benefit from. Hence, the scores we provide are baselines, demonstrating evidence of the feasibility and difficulty of the task.
>
> We will add the discussion around the real-world application of our approach to the camera-ready version of the paper.
>
> -------------------------------------------------
> [1] https://www.redditinc.com/policies/content-policy
>
> [2] https://transparency.fb.com/policies/community-standards/

---

### Official Review · Reviewer_N2ar · 2023-08-05

**Soundness:** 4

**Excitement:**

4: Strong: This paper deepens the understanding of some phenomenon or lowers the barriers to an existing research direction.

**Missing References:**

Hunston, Susan  & Thompson, Geoffrey. 2000. Evaluation in Text : Authorial Stance and the Construction of Discourse: Authorial Stance and the Construction of Discourse. Oxford University Press.

**Paper Topic And Main Contributions:**

The paper is concerned with the analysis and transparency of moderation process discussions in multilingual  online media, in this case Wikipedia which makes these discussions publicly available, albeit not for all such discussions and not always in their full breadth. The aim is an approach for analyzing not only the  moderation itself, but also the stated reasoning behind e.g. suggestions for deletion of content. In presenting an approach to help contributors better understand and transparently explain moderation decisions the approach presented contributes to making moderation decisions more transparent to the community thus potentially boosting user engagement in content creation and moderation.

**Questions For The Authors:**

How do you define stance? What expressions of stance are the most significant features distinguishing moderation statements within your dataset?

**Reasons To Accept:**

The research addresses an important issue, i.e. contributing to the transparency of moderation discussions on online content platforms such as Wikipedia based on the stance of expressed in moderation statements as well we policies issued by the platforms. It takes into consideration data from English as well as German and Turkish, thus addressing the issue for multiple languages even though for the latter two languages there is a data sparsity problem as well as differences between the language versions of Wikipedia as far as explicit policy statements are concerned.

**Reasons To Reject:**

The authors do not address the linguistic aspects of expressions of stance. There is a rich literature on stance in linguistics by scholars like Geoff Thompson, Susan Hunston and Douglas Biber. This literature is not addressed here at all and would potentially contribute to a better understanding of expressions of stance and the respective feature engineering.

**Reproducibility:**

4: Could mostly reproduce the results, but there may be some variation because of sample variance or minor variations in their interpretation of the protocol or method.

**Reviewer Confidence:**

3: Pretty sure, but there's a chance I missed something. Although I have a good feel for this area in general, I did not carefully check the paper's details, e.g., the math, experimental design, or novelty.

---

> ### Author Rebuttal · Authors · 2023-08-28
>
> We thank the reviewer for raising the important point of discussing the linguistic perspective on stance. We define stance along the commonly adopted definition in the field of natural language processing, as the expression of a speaker’s standpoint and judgment towards a given proposition (Biber & Finegan (1988)). In our case, the proposition is the deletion of a given article, and the standpoint is expressed through one of *keep, delete, merge*, or *comment* labels that the author articulates at the beginning of their comment. We will include the definition of stance as well as the relevant citations in the related work section to provide more context on the area as a whole.
>
> To measure common features in the comments expressing stance, we trained a TF-IDF based linear classifier to understand the words most associated with each of the four stances expressed in the comments. We will add a visualisation of our findings to the paper. Below, we provide a textual version of the bi-grams with the highest and lowest weights for the above classifier. We can see that negative expressions like “fails and” and “not enough” are very highly weighted with the *delete* class and negatively weighted for the *keep* class, which makes intuitive sense. Similarly, "clearly passes" and "easily passes", which are presumably highlighting a policy that the article fulfills are associated with the *keep* class, further demonstrating the intertwined nature of editor stance and policy references. The bi-gram “per nom”, an abbreviation for "per nominator" or “per nomination”, has positive weights for the *delete* and *merge* classes while negative ones for the other two, indicating that when the article’s nomination for deletion is referenced, the article tends to be voted for deletion or merging. “can you” being the highest weighted bi-gram for *comment* is intuitive as well since a request for a change is being made.
> | target   | feature                |    weight |
> |:---------|:-----------------------|----------:|
> | delete   | fails and              |   9.08073 |
> | delete   | per nom                |   7.95096 |
> | delete   | non notable            |   6.68485 |
> | delete   | no reliable            |   6.26482 |
> | delete   | not enough             |   6.09139 |
> | delete   | no sources             |   5.32827 |
> | delete   | fails the              |   5.23198 |
> | delete   | not notable            |   5.13093 |
> | keep     | clearly passes         |   6.90419 |
> | keep     | easily passes          |   6.32113 |
> | keep     | clearly notable        |   6.09736 |
> | keep     | not notable            |  -6.11826 |
> | keep     | non notable            |  -6.23591 |
> | keep     | no reliable            |  -6.27908 |
> | keep     | per nom                |  -7.2516  |
> | keep     | not enough             |  -9.10034 |
> | keep     | fails and              | -10.0099  |
> | merge    | to list                |   9.22478 ||
> | merge    | search term            |   6.79893 |
> | merge    | per nom                |   5.69042 |
> | merge    | standalone article     |   5.48076 |
> | merge    | as suggested           |   5.23025 |
> | merge    | to the                 |   5.03847 |
> | comment  | can you                |   4.74349 |
> | comment  | my nomination          |   4.4577  |
> | comment  | this discussion        |   3.98882 |
> | comment  | not sure               |   3.9844  |
> | comment  | this afd               |   3.97706 |
> | comment  | per nom                |  -6.38978 |
>
> --------------------------------------------------------------------------------
> Biber, D., & Finegan, E. (1988). Adverbial stance types in English. *Discourse Processes, 11*, 1-34.

---

### Official Review · Reviewer_6EKo · 2023-08-06

**Soundness:** 4

**Excitement:**

4: Strong: This paper deepens the understanding of some phenomenon or lowers the barriers to an existing research direction.

**Missing References:**

Who Did What: Editor Role Identification in Wikipedia, Yang et. al. in ICWSM 2016.
Annotating Social Acts: Authority Claims and Alignment Moves in Wikipedia Talk Pages, Bender et. al. 2011

**Paper Topic And Main Contributions:**

The paper proposes a dataset for action/policy detection and post stance detection during wikipedia editor discussions. The study is motivated by the fact that during editor discussions, the suggested action/policy is rarely explicitly mentioned, especially in non-English wikipedia. The paper introduces two different tasks, policy detection and stance detection, as well as a joint setup, namely transparent stance detection, where the policy and stance are predicted jointly.

**Questions For The Authors:**

It would be cool to include some error analysis, e.g. from the best baseline system, so that people who work on your task in the future would understand where to make improvements.

**Reasons To Accept:**

The task of policy and stance detection is well motivated and has real world implications.
The dataset + task design is well-aligned with the real world use case of making editorial decisions on Wikipedia.
The proposed multilingual dataset could facilitate future research on the direction, and will be interesting across research community of different languages.

**Reasons To Reject:**

IMHO Ideally the paper should feature some analysis on the quality of the test set in the dataset. e.g. Check human / expert agreement compared to the labels in the test/dev set on a small sample. This can give us a sense of what "oracle" or upper-bound performance on your test set, which will help contextualize the performance numbers you get out of your baselines.

Happy to raise my ratings if the authors can address my concerns.

**Reproducibility:**

3: Could reproduce the results with some difficulty. The settings of parameters are underspecified or subjectively determined; the training/evaluation data are not widely available.

**Reviewer Confidence:**

4: Quite sure. I tried to check the important points carefully. It's unlikely, though conceivable, that I missed something that should affect my ratings.

**Typos Grammar Style And Presentation Improvements:**

Line 908, 990, 1061, etc -- In appendix: Remove all mentions of "Any other comments?" :P?

---

> ### Author Rebuttal · Authors · 2023-08-28
>
> We thank the reviewer for their insightful feedback.
>
> ### Human Annotation
>
> Per the reviewers’ suggestion, we sampled a subset of 100 randomly selected comments from the test dataset and did a preliminary human annotation study on this subset, for both stance and policy prediction. The accuracy scores between our annotation and the labels (which were self annotated by the Wikipedia editors) are 88% and 55% for stance and policy prediction respectively. We attribute the relatively lower score for policy prediction to the large number of labels (94), i.e., policies, with a random prediction baseline of 1%. These policies also at times overlap with each other e.g. *Wikipedia:Lists* and *Wikipedia:Categories, lists, and navigation templates*. While we merged some of the policies with the highest overlap (e.g. notability criteria for specific article types were merged into the *Wikipedia:Notability* policy), for a realistic setup we kept the policies with lesser overlap separate. We plan to extend the human annotation experiment to a larger sample, given the time constraint we wanted to give an insight into the possible outcome of the human annotation study already.
>
> ### Error Analysis
>
> Regarding the proposed error analysis, we performed a brief error analysis of the best model for the English stance detection task i.e. English BERT. Below, we provide the F1 scores for each label, the confusion matrix, as well as examples of comments that the model got wrong with the highest probability. Clearly, the *comment* category is the one the model has the hardest time with, often being mixed up with *keep* or *delete*. This makes intuitive sense as comments conceptually capture a broader range of statements than the other categories. The model has a tendency to over-predict *delete* due to its overrepresentation in the dataset, leaving scope for future studies to address this skewness. Looking at the examples (“Erroneous predictions with highest confidence” below), we can see instances of the most frequent errors (*delete-keep, delete-comment*) made by the model.
>
> ~ F1 score per label ~
> |         |   F1 score |
> |:--------|-----------:|
> | comment |   0.474584 |
> | delete  |   0.944729 |
> | keep    |   0.885506 |
> | merge   |   0.763986 |
>
>
>
> ~ Confusion Matrix ~
>
> |         |   comment |   delete |   keep |   merge |
> |:--------|----------:|---------:|-------:|--------:|
> | comment |      1083 |     1121 |    866 |      22 |
> | delete  |       188 |    27092 |    498 |     112 |
> | keep    |       182 |      795 |   9830 |      51 |
> | merge   |        19 |      456 |    150 |    1311 |
>
> ~ Erroneous predictions with the highest confidence ~
>
> *Please note that policies have been removed, which leads to incomplete sentences such as “fails <space>, and <space>.”, where editors refer to policies in the <space>.*
>
> * Instance 4911: no notablity proven, the references do not mention the subject of the article and only one, at best, is a |reliable source
>   * Predicted Label: delete
>   * Actual Label: comment
> * Instance 11806: and redirect, fails ,  and . The term is a neologism and is not used in a manner to denote fiscal policy, either contemporarily or in archaic terminology. &mdash
>   * Predicted Label: delete
>   * Actual Label: keep
> * Instance 37411: as it appears to exist for the purpose of a |synthetic association between this scene and the TV pilot
>   * Predicted Label: keep
>   * Actual Label: merge
> * Instance 39079: The sources added by voceditenore seem to adequately establish the notability requirements at
>   * Predicted Label: delete
>   * Actual Label: keep
> --------------------------------------
>
> We will add all the points discussed above to the final manuscript.

---

### Meta-Review · Area_Chair_7D5Y · 2023-09-18

**Recommendation:** 4

**Metareview:**

This work presents a dataset of Wikipedia editor discussions for policy detection and stance detection in three languages, along with models for the two tasks and a joint task.

Reviewers appreciated addressing the important issue of content moderation with a focus on transparency. Also, the multilingual dataset and models released will be useful for future research.

The reviewers share a sentiment that a deeper analysis of the dataset from a linguistic perspective can be beneficial, shedding light on linguistic characteristics of different classes and confirming the quality of the dataset in the process. Also, the focus on the Wikipedia editor discussions can be a weakness as it is hard to confirm generalizability of the findings.

---

### Decision · Program_Chairs · 2023-10-07

**Decision:**

Accept-Main

**Comment:**

This work presents a dataset of Wikipedia editor discussions for policy detection and stance detection in three languages, along with models for the two tasks and a joint task.

Reviewers appreciated addressing the important issue of content moderation with a focus on transparency. Also, the multilingual dataset and models released will be useful for future research.

The reviewers share a sentiment that a deeper analysis of the dataset from a linguistic perspective can be beneficial, shedding light on linguistic characteristics of different classes and confirming the quality of the dataset in the process. Also, the focus on the Wikipedia editor discussions can be a weakness as it is hard to confirm generalizability of the findings.